# Simple Method for Apples’ Bruise Area Prediction

**DOI:** 10.3390/ma15010139

**Published:** 2021-12-25

**Authors:** Monika Słupska, Ewa Syguła, Piotr Komarnicki, Wiesław Szulczewski, Roman Stopa

**Affiliations:** 1Institute of Agricultural Engineering, Wrocław University of Environmental and Life Sciences, 37 Chełmońskiego Str., 51-630 Wroclaw, Poland; piotr.komarnicki@upwr.edu.pl (P.K.); roman.stopa@upwr.edu.pl (R.S.); 2Department of Applied Bioeconomy, Wrocław University of Environmental and Life Sciences, 37a Chełmońskiego Str., 51-630 Wroclaw, Poland; ewa.sygula@upwr.edu.pl; 3Department of Mathematics, Wrocław University of Environmental and Life Sciences, ul. Grunwaldzka 53, 50-357 Wroclaw, Poland; wieslaw.szulczewski@upwr.edu.pl

**Keywords:** bruise volume, apple, free fall test, drop test, numerical model, nonlinear estimation

## Abstract

From the producers’ point of view, there is no universal and quick method to predict bruise area when dropping an apple from a certain height onto a certain type of substrate. In this study the authors presented a very simple method to estimate bruise volume based on drop height and substrate material. Three varieties of apples were selected for the study: Idared, Golden Delicious, and Jonagold. Their weight, turgor, moisture, and sugar content were measured to determine morphological differences. In the next step, fruit bruise volumes were determined after a free fall test from a height of 10 to 150 mm in 10 mm increments. Based on the results of the research, linear regression models were performed to predict bruise volume on the basis of the drop height and type of substrate on which the fruit was dropped. Wood and concrete represented the stiffest substrates and it was expected that wood would respond more subtly during the free fall test. Meanwhile, wood appeared to react almost identically to concrete. Corrugated cardboard minimized bruising at the lowest discharge heights, but as the drop height increased, the cardboard degraded and the apple bruising level reached the results as for wood and concrete. Contrary to cardboard, the foam protected apples from bruising up to a drop height of 50 mm and absorbed kinetic energy up to the highest drop heights. Idared proved to be the most resistant to damage, while Golden Delicious was medium and Jonagold was least resistant to damage. Numerical models are a practical tool to quickly estimate bruise volume with an accuracy of about 75% for collective models (including all cultivars dropped on each of the given substrate) and 93% for separate models (including single cultivar dropped on each of the given substrate).

## 1. Introduction

Despite growing environmental awareness, consumers still demand food products with impeccable appearance and condition [1]. Studies have shown that about 30–40% of food is wasted due to bruising [2]. Irreversible damage to the fruit disqualifies the product from use and if the company that owns it does not have a Zero Waste Policy, then the fruit becomes waste, which then goes to landfill or at best is disposed of in bio-waste. The bruising of the fruit is closely related to the variety and stage of ripeness. These in turn can be described by firmness, skin thickness, amount of wax covering it, organic acid, and sugar content [3,4].

Post-harvest processes cause mechanical damage during reloading, packaging, and storage, which are a result of different impacts and quasi-static loads. A large percentage of fruit bruises is caused by improper transport methods causing higher loads than static storage. Most of the damages are caused when one fruit hits another or a stiff surface. Fruit damage is determined by drop height, type of surface against which the fruit falls, or velocity at the moment of impact. Laboratory tests have shown that fruit located on the top layer of a pile exhibits more damage than those located at the bottom layer. This increased damage is caused by a higher level of acceleration in the upper layer [5,6,7]. Mechanical damage is difficult to eliminate because biological material consists of a delicate cell structure. Therefore, it is important to reduce impact intensity by absorbing part of the energy to reduce negative influences on the tested fruit. Minimization results in the distribution of the impact force on greater area of the fruit, and hence reduces pressure on the tissue (force per unit area) and the probability of bruise occurrence. Currently, there are several methods of biological material protection during transport, i.e., through the use of vehicles equipped with additional pneumatic vibration damping elements, as well as lining containers with protective materials to protect fruits during direct contact [8,9,10,11,12].

Current studies on fruits and vegetables bruises are carried out using both, destructive and nondestructive methods. Destructive methods are more common because of lower costs of research, but they are more time-consuming. Nondestructive methods require the use of specialized equipment, which entails high costs, but also the higher efficiency of the tests. In recent years, there has been a greater interest in the non-destructive method, which will most likely replace the methods used so far. Opinions about the precision of both methods are still divided [13,14,15,16,17,18,19].

From the existing literature, it follows that susceptibility or resistance to bruises is usually described by the applied energy and the resulting bruise volume [2,20]. The existing literature and the authors’ own experience have shown that indicators based on the strength properties of substrate material are not a less important parameter. On the basis of these circumstances, the maximum surface pressure transferred by apple tissues without triggering further damage, bruise resistance, and bruise level can be distinguished [21,22]. Research indicated the difficulty in determining bruise volumes, which are caused by irregular bruise shapes in fruit flesh [23]. The bruise threshold is described as a permissible drop height, at which the fruit bruise is specified by a threshold determined under conditions similar to natural loads, especially in relation to velocity and impact energy. Hence, many studies were conducted for measuring impact loads [13,24,25]. A comparison of changes in contact during impact at four different substrates has shown how the material of substrates influences bruise levels [26,27]. Research has also indicated that the bruise surface is a better parameter to assess the bruise of fresh fruit than bruise volume [28]. 

So far, most of the research conducted in this discipline has mainly focused on implementing new methods for determining the volume of bruises. Nevertheless, from the producers’ point of view, there is no universal and quick method to predict losses when higher than standard forces are applied, for example when dropping a product from a certain height onto a certain type of substrate. Such information is also crucial for manufacturers of technological lines for a specific product. In this study the authors presented a simple method to estimate bruise volume based on drop height and substrate material. The model was supported by empirical results from free drop tests.

## 2. Materials and Methods

### 2.1. Characteristics and Preparation of Research Material

These studies were performed on three apple cultivars—Jonagold, Golden Delicious, and Idared—with origin from the Lower Silesia Voivodship in Poland. The fruit was purchased directly from the producer and stored for nearly one month after harvest in a storeroom equipped with a ULO (Ultra Low Oxygen) system. The fruit was stored under controlled atmosphere at 2 °C, 95% relative humidity, 0.7% carbon dioxide, and 2% oxygen. Table 1 presents the basic characteristics of the tested fruit.

The tested fruit was carefully sorted according to its geometrical characteristics: the mean diameter and mass. Twenty fruits were randomly selected from each cultivar, and then the firmness was measured using a hand-held tester with a diameter of 11.1 mm (Facchini FT 327, Alfonsine, Italy). The firmness test was performed on the tissue after peel removal with a special knife. In addition, the fruits’ water content was determined using the weight-dryer method, while the sugar content of the fruit was measured using refractometer (RMR 200 hand refractometer, Hanna Instruments, Woonsocket, Rhode Island, USA). The sugar measurement range was 0–32%, and the accuracy range was within ±0.1°.

### 2.2. Free Fall Test

For the free fall test, a special device was used to drop the fruit from a specified height ranging from 10 to 150 mm (Figure 1). Apples were dropped onto four substrates: concrete 20 mm thick, spruce panel 5 mm thick, corrugated cardboard 5 mm thick (500 g/m^2^), and polyethylene foam 3 mm thick. The moduli of elasticity of substrates were as follows: concrete 28.5 GPa, wood 10.4 GPa, corrugated cardboard 1.1 GPa, and foam 0.05 GPa. The total number of tested apples was 900. Details of the measuring device and the measurement methodology are described in another article [27].

### 2.3. Determining the Surface of the Upholstery on the Basis of Image Analysis

After the impact tests, the fruit were quarantined for 4 days at a temperature of 25 °C. Then, for better image analysis, the peels around the bruises were removed (Figure 2a). Based on the images taken by the Nikon D3100 camera, computerized image analysis of the bruised surface area was performed. The images were contrasted in ImageJ software. Then, the images were scaled in AutoCAD and the bruises perimeters were determined using the SPLINE command (Figure 2b). Next, using the LIST command, the bruises areas were determined for all tested fruits (Figure 2c).

### 2.4. Linear Model

The simplest linear model was proposed to describe the phenomenon of fruit bruise occurrence under certain drop conditions. The following equation is an explanation of the linear function interpretation.
(1)B=a·H+b

B—bruise area, mm^2^H—drop height, mma—slope,b—intercept.

In the proposed linear model, the dependent variable determining the fruit bruise area was determined from the independent variable determining the drop height. It should be noted that only positive values of the slope indicate the positive linear correlation that exists in the analyzed data. 

For the data analyzed, the intercept affects the volume of bruise occurring at initial drop heights. When there is a positive intercept value, the initial drop heights will generate bruises on the fruit. On the other hand, the negative value of the intercept indicates the lack of bruising of the fruit at the initial drop heights. If X_0_ is negative, the model assumes the occurrence of bruises at the lowest assumed drop height, which is usually taken to be 10 mm. If X_0_ is positive, the value of this indicator is directly proportional to the height at which the bruise occurs. All relations described above are marked in Figure 3. 

In order to evaluate the quality of the proposed linear models, the coefficient of determination R^2^ was determined.

## 3. Results

Figure 4 shows the correlation between bruise area as a function of drop height and variety of apple.

The least resistant to damage was the Jonagold variety, characterized by a high sugar content and lowest firmness. The Golden Delicious variety with a lower sugar content and characterized by medium firmness was more resistant to damage than Jonagold. Meanwhile, due to the lowest sugar content and highest firmness, the Idared variety proved to be the most resistant to damage manifested by bruise area. The above graph is only the background of the study and indicates the reason for the differences between the tested varieties.

Figure 5 shows the correlation between bruise area as a function of drop height and the substrate on which the fruit was dropped.

The graphical distribution of the test results (averaged over all the varieties tested) shows a distribution of results with an increasing tendency for the bruise area with increasing drop height and substrate type according to the following order of growth: foam, cardboard, wood, and concrete. The above presented results were an inspiration for the statistical models based on which it would be possible to determine the size of bruises for a particular cultivar and substrate. The models are presented later in the article.

Appendix A shows the averaged test results and the linear models based on Equation (1) for all varieties. The results were segregated according to the apple variety and the type of substrate the fruit was dropped on. The models fit values (R^2^) along with the *a* coefficients and y-intercept for each substrate type are included further in Table 1, summarizing all models of apple varieties. From these models, it can be concluded that such separate models (including single variety dropped on each of the given substrate) can be used to quickly estimate bruise volume with an accuracy of about 93% while dropping fruit from a particular height.

Figure 6 shows the mean measurement results of the three cultivars analyzed and their combined models for each substrate. The graph not only captures the differences between varieties—with Idared showing the lowest bruising volume values, Golden Delicious the medium, and Jonagold the highest, but also captures the similarity in fruit behavior on different substrates. From this figure, it can be concluded that such a combined model (including all cultivars dropped on each of the given substrate) can be used to quickly estimate bruise volume with an accuracy of about 75% while dropping fruit from a particular height.

Table 2 below shows the fit values of the individual models and the combined models as well as the values of the slope, intercept, limit value indicator, and model angle. On the basis of the presented table, a spreadsheet (Appendix A) was created, which can be used to apply the results of the research to compare model data.

αααα Based on the results of concrete and wood from Figure 7 it can be concluded that above a certain stiffness of the substrate, there is little difference between the bruise area, as the fruit will absorb most of the kinetic energy. Cardboard, due to its corrugated structure, at low drop heights partially absorbs kinetic energy until its structure is destroyed. Then the energy starts to be absorbed by the concrete located underneath. The foam, at all drop heights considered in this study, absorbs most of the kinetic energy due to the lack of destruction of its structure. For low drop heights of up to 40 mm, the foam does not cause any bruising, while the increase in bruise size, determined by the *a* parameter, is not as high as with cardboard.

The number of observations of the occurrence of apple bruise area on a particular substrate was grouped, as shown in Figure 8. 

For foam, observations occur up to 375 mm^2^ with the largest amount of data (15 observations) occurring below 25 mm^2^. This was due to the low elastic modulus of the foam, which absorbed most of the kinetic energy during free fall. The occurrence of apple bruising for the cardboard substrate ranged from 25 to 525 mm^2^, where 275 mm^2^ was the most common bruise. Compared to the rest of the substrates, cardboard gave the most averaged results relative to the others. For the lowest drop heights, the kinetic energy of the fruit was absorbed by the cardboard and bruises showed lower values. On the other hand, at higher drop heights, the crushed cardboard absorbed less energy, which resulted in a similar behaviour to wood. The size of fruit bruises falling on the wood substrate ranged from 75 to 625 mm^2^, where 325 and 375 mm^2^ were the most common bruises. For the concrete substrate, apple bruising occurred from 125 to 625 mm^2^. Both of these substrates showed great similarity to each other. The size of the bruise area was larger, even at low free fall heights.

It should be noted that bruising below 25 mm^2^ occurred only for foam, while bruising between 25 and 75 mm^2^ occurred for two materials—foam and cardboard. All materials resulted in a bruise size of 125 to 375 mm^2^. On the other hand, the 425 to 525 mm^2^ bruising occurred for three substrates—concrete, wood, and cardboard. The highest bruising above 525 mm^2^ to 625 mm^2^ occurred only for concrete and wood substrates.

## 4. Discussion

The results from Table 1 for sugar content and firmness of the cultivars included in the study were validated against other sources [29] and extended with conclusions on the effect on the mechanical properties of the fruit. Confirmation of the validity of the conclusions regarding the aspects of fruit strength properties can be found in publications of other authors [30], where the issues of SSC, starch, pectin, acids, firmness, and mechanical properties of fruit were studied.

No article has been found on bruise volume in relation to the type of substrate onto which the apples were dropped. Most of the articles are about the correlation between the maturity of a particular variety and bruise volume after a free fall test [31,32,33]. Also, the relation between mass and drop height at impact causing the bruising of apple was investigated by other researchers [34].

Therefore, in an attempt to compare these studies, it can be concluded that just as the stage of fruit ripeness in the aforementioned studies affects the mechanical properties of the fruit, the variety and corresponding biochemical properties are also important. Resulting in bruising, flesh destruction is inhibited by organic acids. Some varieties have more of these acids and some do not. As the fruit ripens, these acids take part in enzymatic reactions and thus their content gradually decreases. Therefore, a fresh, sweeter, and acid-poor variety will be close to the endurance characteristics of a strongly mature but acidic and sugar-poor variety. 

## 5. Conclusions

A fresh, sweeter, and acid-poor variety will be close to the endurance characteristics of a strongly mature but acidic and sugar-poor variety.Separate models (including a single variety dropped on each of the given substrate) can be used to quickly estimate bruise volume with an accuracy of about 93% while dropping fruit from the particular height. However, combined models can be used with an accuracy of about 75%.Above a certain stiffness of the substrate, there is little difference between bruise areas, as the fruit absorbs most of the kinetic energy.Compared to the rest of the substrates, cardboard gave the most average results relative to the others. For low drop heights, it behaved more like foam, and for high drop heights, more like wood.

## Figures and Tables

**Figure 1 materials-15-00139-f001:**
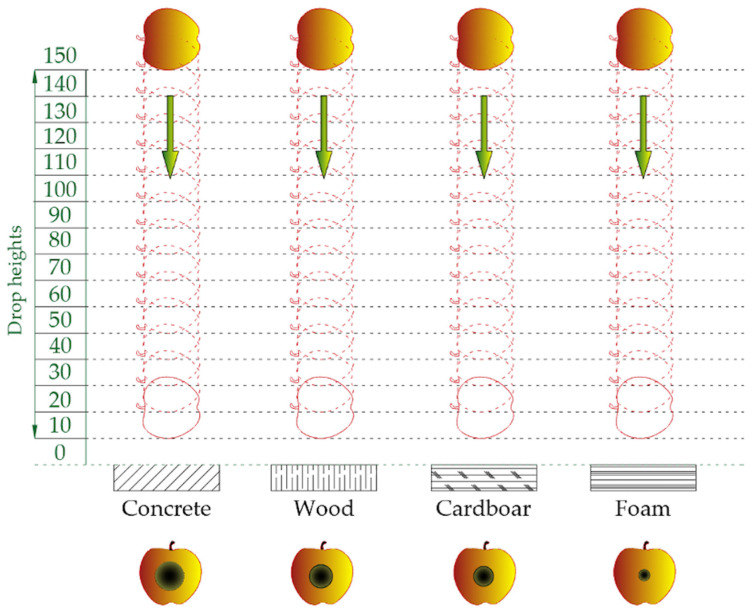
Scheme of free fall test procedure.

**Figure 2 materials-15-00139-f002:**
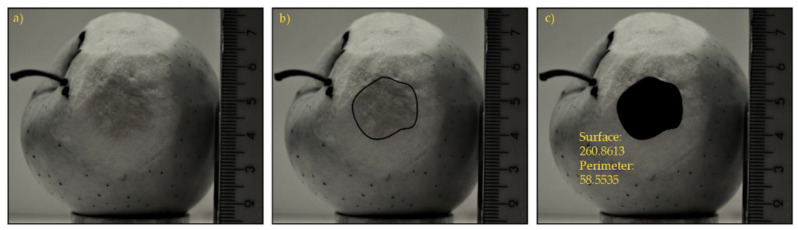
Procedure for image processing and determination of a bruise surface area: (**a**) peel removed; (**b**) bruise perimeters determination; (**c**) bruise area determination.

**Figure 3 materials-15-00139-f003:**
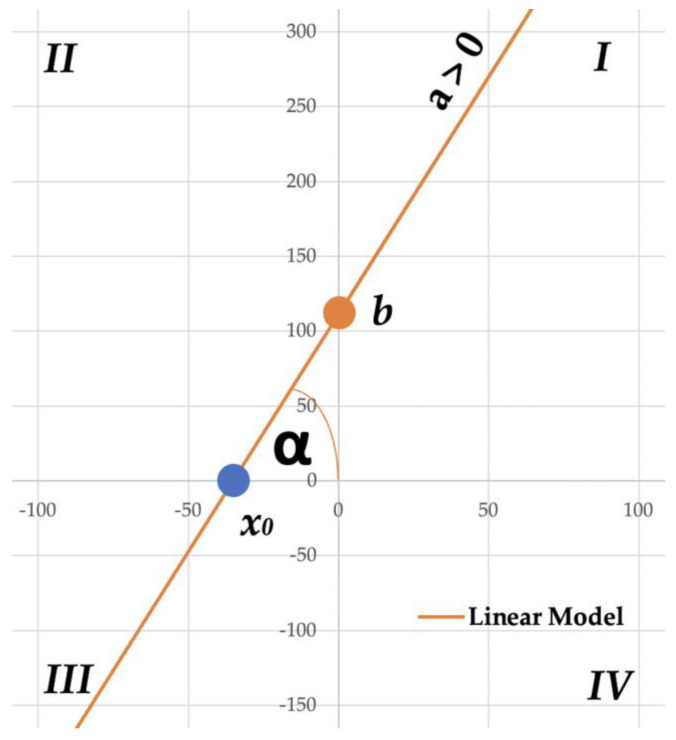
Method of slope, intercept, limit value indicator, and model angle determination.

**Figure 4 materials-15-00139-f004:**
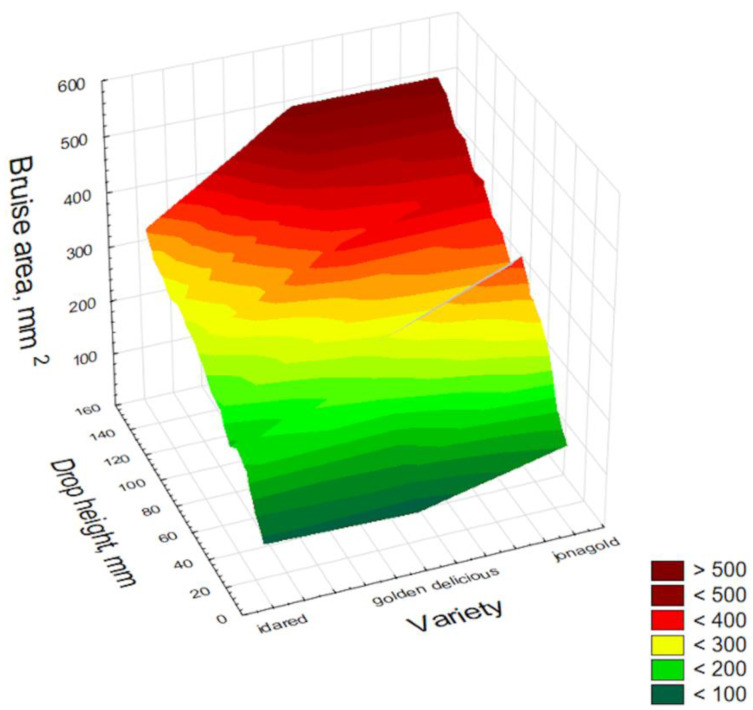
Distribution of apple bruise size in relation to drop height for different apple varieties.

**Figure 5 materials-15-00139-f005:**
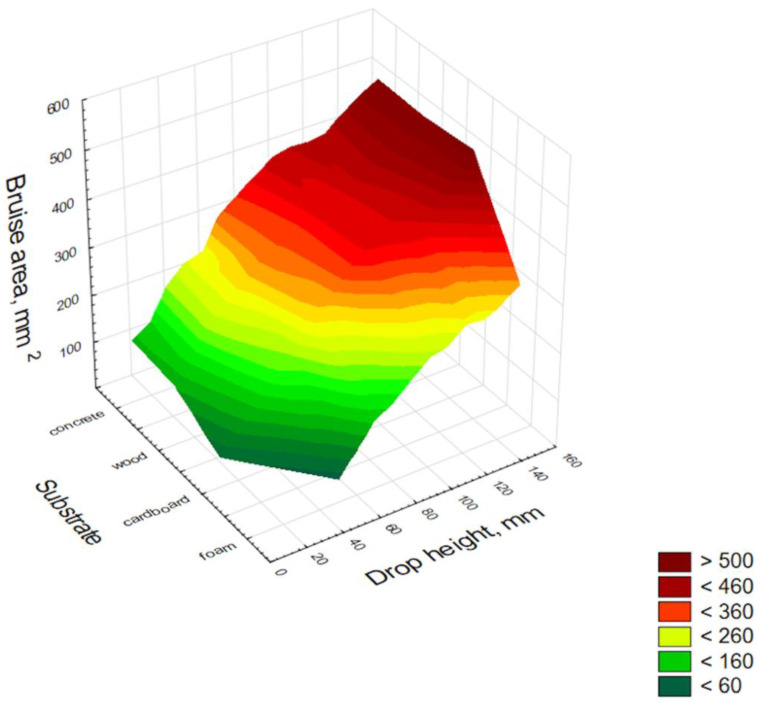
Distribution of apple bruise size as a function of drop height for different substrates.

**Figure 6 materials-15-00139-f006:**
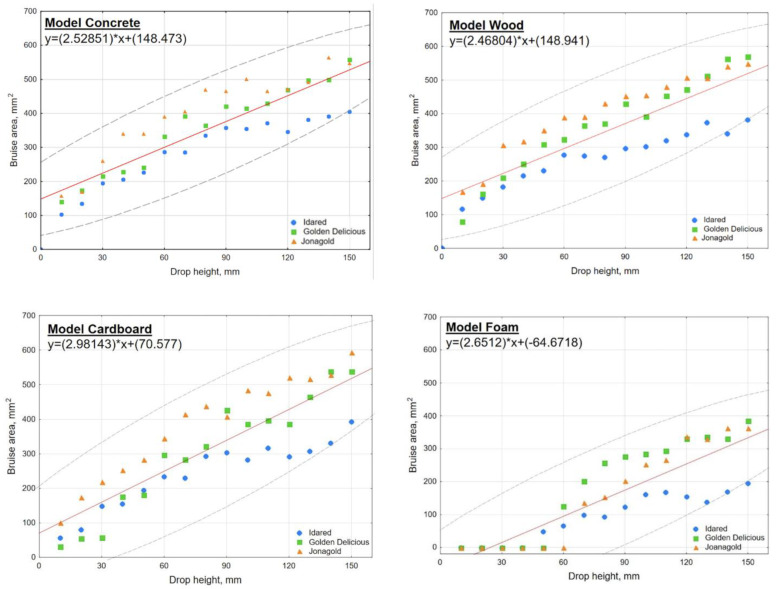
Determination of substrates models based on averaged test results of apple bruise area as a function of drop height.

**Figure 7 materials-15-00139-f007:**
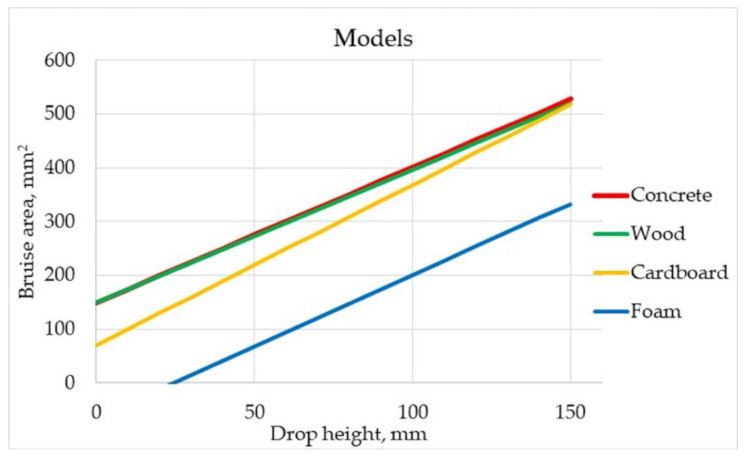
Substrates combined models.

**Figure 8 materials-15-00139-f008:**
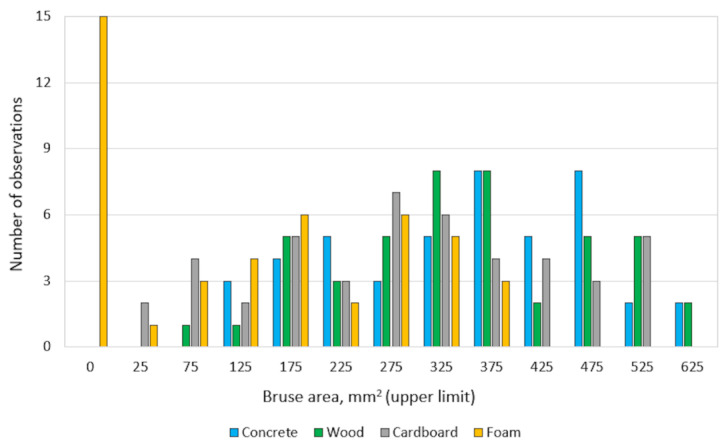
Quantitative analysis of fruit bruise volume for different substrates.

**Table 1 materials-15-00139-t001:** Characteristics of the tested material (presented values described as mean ± standard deviation).

Cultivar	Weight	Mean Diameter	Firmness	Water Content	Sugar Content
–	kg	mm	N	%	%
Jonagold	0.171 ± 0.005	69.1 ± 2.3	63 ± 1.8	84.8 ± 0.5	14.3 ± 1.1
Golden Delicious	0.200 ± 0.004	74.3 ± 1.9	67.4 ± 4.6	87.8 ± 0.8	12.4 ± 0.8
Idared	0.210 ± 0.006	77.6 ± 2.5	75.4 ± 4.8	87.9 ± 1.1	11.8 ± 1.0

**Table 2 materials-15-00139-t002:** Determination of the actual distributions of the bruise area as a function of impact height.

		Idared Model	Golden Delicious Model	Jonagold Model	Combined Model
**Concrete**	a	2.07	2.86	2.66	2.53
	b	126.22	129.59	189.61	148.47
	R^2^	0.91	0.96	0.88	0.78
	α	64	71	69	68
	X_0_	−60.98	−45.15	−71.28	−58,68
**Wood**	a	1.71	3.17	2.53	2.47
	b	135.58	111.46	199.79	148.94
	R^2^	0.94	0.93	0.93	0.72
	α	60	72	68	68
	X_0_	−79.29	−35.16	−78.97	−60.30
**Cardboard**	a	2.05	3.68	3.21	2.98
	b	77.08	8.71	125.95	70.58
	R^2^	0.90	0.94	0.95	0.75
	α	64	75	73	71
	X_0_	−37.60	−2.37	−39.24	−23.68
**Foam**	a	1.52	3.20	3.24	2.65
	b	−27.45	−67.49	−99.08	−64.67
	R^2^	0.92	0.91	0.93	0.77
	α	57	73	73	69
	X_0_	18.06	21.09	30.58	24.40

## Data Availability

Not applicable.

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
