# Peer review of "Simple Method for Apples’ Bruise Area Prediction"

_materials, 2021, doi:10.3390/ma15010139_

Round 1
Reviewer 1 Report
The title of the paper is interesting but the content needs thorough improvement. The specific comments are as follows:
- Line 36: Irreparable..the right term used is irreversible
- Lines 38-40: The bruising of.... Please improve the sentence as it is difficult to understand in the current form
- Line 45: "on a hard surface" what is the meaning of hard surface here?
- Line 54: while the paragraph discussing the general fruit, why suddenly the word "apple" appeared?
- Subtopic 2.1: explain why the study was conducted on the stored fruit and not on fresh-harvested fruit?
- Line 106 - replace "as well as" with "while"
- Line 106 - ...sugar content of the fruit "was measured" using refractometer..
- Lines 111-112 - improve the explanation
- Line 115 - its quite confusing. Line 102 said 20 samples/cultivar but the total number here is 900. Please explain
- Line 126: "A short graphic description.... " please improve the sentence. Difficult to understand
- Section 2.4 - please remove. The statements are repetitions
- Line 140: please improve the sentence
- Equation 1 - where's the eqn comes from??
- section 2.5.2 - the explanation in this section is a general knowledge of mathematics
- The data analysis is not convincing
Author Response
Dear reviewer, we sincerely thank you for your valuable comments. We have attached our responses to the review below.
- Changed to: Irreversible
- Lines 38-40: Improved:
The bruising of the fruit is mainly caused by variety and stage of ripeness. These in turn can be described by firmness, skin thickness, amount of wax covering it, organic acid and sugar content [3,4].
- Line 45: Changed to: stiff surface
- Line 54: "apple" deleted
- Subtopic 2.1: We had surplus of research material, kept in ULO after another research. We knew, after previous impact study, that after good quality ULO storage, fruit will not differ significantly from fruit picked straight from the tree.
- Line 106: Done
- Line 106: Added
- Line 111-112: I'm very sorry, but I don't understand how to improve this explanation. I wrote the description of substrates on which the apples were dropped. I included their thickness, moduli of elasticity and density for the corrugated cardboard. I asked native speaker and he told me that it was correct and very factual form of the description.
- The number of samples in line 102 relates to the characteristics of the test material, while line 115 relates to tests in the free fall test.
- I deleted this redundant sentence and left the reference to Figure 2c in parentheses.
- I fully agree
- Sentence has been improved
- The following equation is an explanation of the linear function interpretation. We improved description of the Eqn in the text.
- The entire section (Linear model) has been revised
- Data analysis was revised as recommended by the second reviewer. Anyway, if data analysis is still not convincing, could you kindly suggest what should be improved.
In table attached in pdf the last sentence in point 15 is redundant. Sorry for our mistake, we couldn't withdraw that file.
Best regards.

Reviewer 2 Report
Dear Authors,
Thank you for this really nice work you exposed in your manuscript. A really good job and I enjoyed reading it. However, I cannot accept your manuscript in present form. First, please split the chapter 3. Results and discussion into two chapters.
- Precisely, in Chapter 3: Results, please just explain the results you achieved by using the research methods you explained very nice in chapter 2. Materials and methods, and, in Chapter 4: Discussion, please just write the comments about results exposed in previous chapter. Moreover, I strictly require from you to write one more chapter, in that new Chapter 5: Conclusion, please just the most important topics of your research which are of huge importance for every apple producer, or organizer of packaging in ULO storage, and not to mention the other group of readers academicians and students, etc.
- Nevertheless, considering the first paragraph in Chapter 2. Cit. “These studies were performed on the three apple cultivars Jonagold, Golden Delicious, and Idared, with an origin from the Lower Silesia Voivodship in Poland. The fruit 92 was purchased directly from the producer and was stored for nearly one month after har-93 vest in a storeroom equipped with a ULO (Ultra Low Oxygen) system. The fruit was 94 stored under controlled atmosphere at 2°C, 95% relative humidity, 0.7% carbon dioxide, 95 and 2% oxygen.”, I strongly suggest you to continue this research in the future and to take in consideration the different time of storage in ULO, and also some other characteristics of apple fruits, such as water content and sugar content. Also, I suggest you to use more apple cultivars in future research. This research and obtained results you exposed in this manuscript, could be an excellent start point for other investigation and you will also have a nice reference for citation(s) in future manuscript and papers you will publish.
- Oh, I almost forgot, please in table 1. You used the inappropriate term “Humidity” and I strongly suggest you to replace it with the term “Water content”, because you have a water content in the apple fruit and when you use the term “Humidity” first association for every reader is air humidity in the storage. Then please separate the units and degrees and percent from its numerical values with one click on spacebar, etc.
Conclusion: My congratulations for your very nice work and I will accept your manuscript for publishing after some major improvements I requested under the point 1, and after you will do some technical corrections in text I requested under the point 3. and please, consider my suggestion I exposed under the point 3.
Thank you
Author Response
Dear reviewer,
we sincerely thank you for your valuable comments. We have attached our responses to the review below.
1. Done
2. We are currently doing such a project :)
3. Done
We greatly appreciate your valuable time you took to review. Thank you also for your positive comments, which give us positive energy to continue our work.
Best regards.
Round 2
Reviewer 2 Report
My congratulations, this is exactly what I meant. You really improved your manuscript according to my recommendations. It is a very nice paper now; it is more readable and perceptive for readers, which is very important because your work is very interesting for practitioners in apple post-harvest management. Reviewing your manuscript was a great pleasure for me. Thank you. Oh, I almost forgot, I wish to congratulate to colleague(s) who defended their doctoral dissertation(s) with such nice, applicable and moreover, comprehensive research. I wish you all the best my young colleague(s). Thank you for some joy in such grey, wet and worrying time. :-)